# BMJ Open | Do we need a core curriculum for medical students? A scoping review

Maulina Sharma,[1,2] Ruth Murphy,[3] Gillian A Doody[1]

[1]Medical Education Centre, School of Medicine, University of Nottingham, Nottingham, UK
[2]Dermatology, University Hospitals of Derby & Burton NHS Foundation Trust, Derby, UK
[3]School of Medicine, University of Nottingham, Nottingham, UK

**Correspondence to**
Dr Maulina Sharma;
msxms16@nottingham.ac.uk

## ABSTRACT

**Objective** The General Medical Council (GMC) recommends medical schools to develop and implement curricula enabling students to achieve the required learning outcomes. UK medical schools follow the GMC's Outcomes for graduates, which are generic. GMC plans to introduce a national Medical Licensing Assessment (MLA) for the medical graduates wanting to practise medicine in the UK in 2022. With no standardised or unified undergraduate (UG) curriculum in UK, various specialties have expressed concerns about not being represented in medical schools and developed specialty-specific core curricula. The aim of this review was to identify learned bodies who have developed a core curriculum for UK medical schools and highlight the drivers, gaps and future approaches to curricular development and implementation.

**Methods** A literature search was conducted using online databases (EMBASE, MEDLINE, ERIC, HMIC, PubMed and CDSR), search engines and related websites (Google and Google Scholar, Department of Health, GMC and BMA) for relevant articles from 1996 to 5 March 2019 (~20 years). A methodological framework to map the key concepts of UG medical curriculum was followed. Any relevant body with a core curriculum for UK medical UGs was included.

**Results** A total of 1283 articles were analysed with 31 articles included in the qualitative synthesis, comprising 26 specialties (clinical n=18, foundation subjects n=4 and professionalism related n=4). WHO, European and national (eg, Royal Colleges of UK) specialty bodies provided specific core learning outcomes for the medical graduates. Patient safety, disease burden, needs of society and inadequate preparedness of medical graduates were drivers for the development of these curricula.

**Conclusions** This is the first comprehensive review of literature on UG core curricula recommending minimum standards on knowledge and skills, in alignment with GMC's Outcomes for graduates for all the UK medical students. Adopting and assessing unified standards would help reduce variability across UK medical schools for both generic and specialty-specific competencies.

## INTRODUCTION

The General Medical Council (GMC) set out the Promoting Excellence document in 2016, which outlines a set of standards for all stages of medical education and training, including the undergraduate (UG) medical education.[1] These standards should ensure that medical schools develop and implement curricula to enable students to achieve learning outcomes

---

**Strengths and limitations of this study**

► Scoping review of literature as the methodological approach in the study helped identify, categorise and provide a narrative synthesis of the key aspects on core curricula for UK medical students.
► A strength of the study is that it is the first comprehensive review on core curricula for UK medical students, which could help medical schools in curriculum mapping and implementation.
► The scoping study highlighted gaps in specific undergraduate curricula as well as the heterogeneity of the curricular designs between specialties and subjects.
► A limitation of the study is that heterogeneity in the curricular designs for specialties made comparisons between them difficult.
► As with limitations seen in other scoping reviews, articles or studies pertinent to the review question may have been unintentionally omitted.

---

required as graduates and practise safely and competently. The GMC also plans to introduce a Medical Licensing Assessment (MLA) in 2022 applicable to medical graduates who want to practise medicine in the UK.[2] Specialties have expressed concern about medical curricula as medical schools in the UK currently organise their own curricula and assessments based on GMC's Outcomes for graduates, which can be viewed as generic.[3] This may result in scanty representation of specialist knowledge and skills within the curriculum. Once the students successfully graduate, the GMC grants them a licence to practise.

In the UK, there is a growing need for the development of general practitioners (GPs), who treat all common medical conditions and refer patients into secondary care for urgent or specialist treatment. Also, a 'Future Hospital Commission' report, through the Royal College of Physicians (RCP) in 2013, promoted the reintroduction of generalism into medical training.[4] This means that medical education and training of doctors in UK needs to address the current and future demographic of patients.

With the generalist trend, as well as plans to introduce the MLA for all medical graduates in the UK, medical schools may be influenced on what they teach and assess. There is currently no unified or standardised UG medical curriculum in the UK. Various specialties have expressed concerns about not being adequately represented in medical school curricula and have developed specialty-specific core curricula.

Our objective was to perform a scoping review of literature to identify the learned bodies who have developed a core curriculum for the UK medical schools and to use this information to highlight the drivers, gaps and future approaches to curricular development and implementation.

## METHODS

We performed a scoping review using the methodological framework as outlined by Arksey and O'Malley.[5] The framework was used to: identify the specialties, or subjects which recommended and had developed an UG core curriculum for the UK medical schools; select and search for relevant studies; chart and collate the data, and summarise the results.

### Search strategy

A literature search was conducted using online databases (EMBASE, MEDLINE, HMIC (searched via NICE Healthcare Databases Advanced Search), PubMed, The Cochrane Library and ERIC (Education Resources Information Centre). Other online search engines (Google and Google Scholar) and websites of Department of Health, GMC and British Medical Association (BMA) were also searched for relevant articles from 1996 to May 2017 (ie, the past 20 years). The literature search was updated up to 5 March 2019 for any further recent relevant articles.

Keywords or terms used in the searched included: GMC, medical student, medical education, UG, curriculum, standards and national (UK and individual countries, eg, England). A draft of the full search strategy is included in online supplementary appendix 1.

Electronic search results were managed using a reference manager. Titles and abstracts of all citations were first screened by the author (MS) and those that were not related to UG curricula or the UK were excluded. Full articles were requested from the library, when it was not possible to eliminate them from reading either their title or abstract. The full text of all potentially eligible articles were then obtained and assessed against the inclusion criteria (MS and RM). Where the full text of an article was not available the authors were contacted to request a copy. Any ambiguities about whether a study met the inclusion criteria were resolved by a consensus between authors (MS, RM and GAD).

### Eligibility criteria

Inclusion criteria were all published studies, reports or articles, which had a recommended core curriculum for a specialty or subject for the UK medical UGs on a national level. Any international or national body, society, college or organisation specific and pertinent to UK medical UG education was included. For curricula which had been revised or updated, the most recent (up to 5 March 2019) were included.

Exclusion criteria were articles pertaining to core curricula for postgraduate courses and those not specific to UK medical UGs. Due to the study being relevant to UK setting, non-English language literature was excluded.

### Patient and public involvement

As this was a scoping review of available UG core curricula, patients or public were not involved.

### Data management

Data were collated and reported using Preferred Reporting Items for Systematic Reviews and Meta-Analyses flow diagram (figure 1). Characteristics of each study were recorded in an electronic data collection table and included the name of specialty or subject, in alphabetical order, organisation or body developing the curriculum, year of the curriculum update, and whether the studies were aligned to GMC's learning outcomes (table 1).

The key concepts derived were broadly grouped into the following categories: grouping of core curricula and their timing within the school course; organisations involved in development of the core curricula; drivers for development of core curricula and curriculum overlaps and gaps.

## RESULTS

The initial search identified a total of 1283 articles, after duplicates were removed (n=21). All articles were screened and 126 were initially highlighted as of interest. Of these, 95 were subsequently excluded, with reasons recorded. A total of 31 articles, published since 2002 and describing core curricula recommendations for UK medical UGs, were then included in the narrative synthesis. These comprised a total of 26 specialties or subjects (figure 1). Of these 26 specialties, five specialties had additional relevant articles (table 1).

### Narrative synthesis of results

Having identified publications from learned bodies that had developed a core curriculum for the UK medical schools, we identified the following themes and drivers for curricular development and implementation, and summarised areas where there were gaps and overlaps which may influence future approaches to curricular development and implementation.

#### Grouping of curricula and timing within the school course

The curricula developed and recommended for UK medical schools could be grouped into three main groups: clinical specialty based (n=18) (eg, surgery and dermatology); foundation subjects (n=4) (eg, anatomy

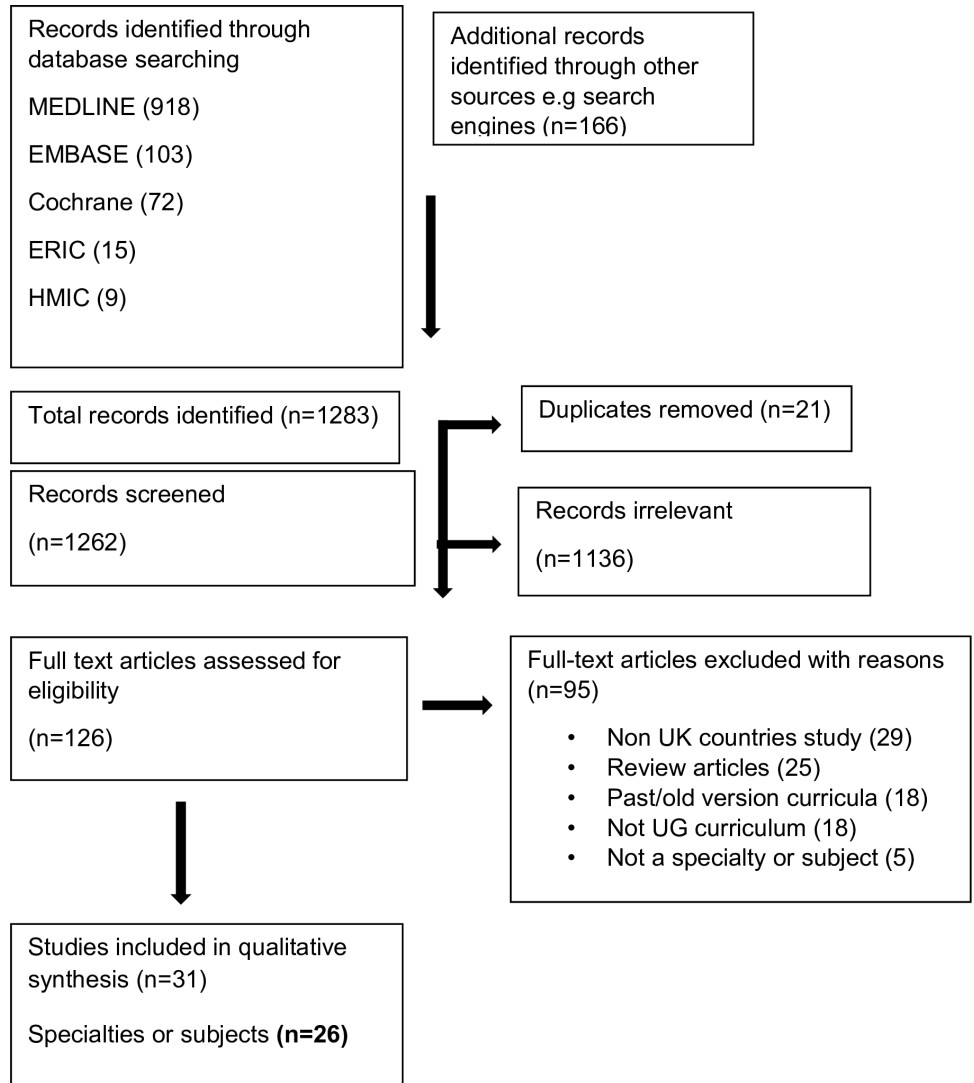

**Figure 1** PRISMA flow diagram of the scoping review. PRISMA, Preferred Reporting Items for Systematic Reviews and Meta-Analyses; UG, undergraduate.

and pharmacology) and subjects related to professionalism (n=4) (eg, communication skills) (table 1).

Clinical specialty curricula were applicable during the clinical years of training, for example, paediatrics, obstetrics and gynaecology, psychiatry, surgery, acute medical care, resuscitation and end-of-life care.[6–12] Some specialties had developed a curriculum with a system-based approach, for example, neurology and dermatology.[13 14] Foundation subjects like anatomy, pathology and nutrition as well as curricula related to professionalism, such as medical ethics and law, were developed so as to be relevant throughout all medical school training.[15–18]

### Organisations involved in development of individual core curricula

Curricula were developed by international, national and regional organisations with participation from relevant stakeholders. Consensus agreement derived from a Delphi process was the preferred method of achieving core curricular recommendations.[10 11 15 18–21]

Four curricula namely communications skills, geriatrics, musculoskeletal system and palliative care had their core curriculum codeveloped by two different international and national bodies. The international recommendations encompassed the national guidance.[19–24]

The Royal Colleges of the UK developed specialty-specific core curricula for nine specialties (eg, general practice, pathology, radiology and surgery).[6–9 16 25–28]

National specialist consultant societies and bodies developed 14 specialty-specific curricula (eg, dermatology, medical ethics and palliative care).[10–15 17 18 21–24 29–31]

National expert consensus through a survey of consultants was used to develop guidelines for core curricula in urology and ENT, while a separate urology syllabus for medical UGs was developed by the British Association of Urological Surgeons.[32–34] At a regional level, one medical school developed and recommended a curriculum for medical humanities.[35]

**Table 1** Electronic data collection table of specialty or subject with curriculum in alphabetical order

| | Specialty with curriculum | Year | Organisation/body | Alignment with GMC outcomes |
|---|---|---|---|---|
| 1. | Acute care[10] | 2005 | Resuscitation Council UK | Y |
| 2. | Regional anatomy[15] | 2016 | Anatomical Society | Y |
| 3. | Clinical pharmacology and prescribing[30] | 2012 | British Pharmaceutical Society | Y |
| 4. A B | Core communications curriculum[20] Communications curricula[21] | 2013 2018 | European study using Delphi process UK Council for Clinical Communication Skills Teaching in Undergraduate Medical Education | Y |
| 5. | Dermatology[14] | 2016 | British association of dermatologists | Y |
| 6. | ENT[34] | 2014 | A Delphi survey (UK) of ENT consultants and specialist registrars, accident and emergency consultants and specialist registrars, general practitioners and paediatricians. | Y |
| 7. | General practice[25] | 2018 | Royal College of General Practitioners | Y |
| 8. A B | European undergraduate curriculum in geriatric medicine[19] Geriatric medicine for undergraduates[22] | 2014 2013 | European Union of Medical Specialists British Geriatrics Society | Y Y |
| 9. | Medical ethics and law[18] | 2010 | Medical Education Working Group of the Institute of Medical Ethics and associated signatories | Y |
| 10. | Medical humanities[35] | 2006 | Peninsula Medical School | Unclear |
| 11.A B | Musculoskeletal system[23] Regional Examination of the Musculoskeletal System[24] | 2004 2004 | The Bone and Joint Decade Undergraduate Curriculum Development Group UK representation of the four specialties- Rheumatologists, Orthopaedic Surgeons, Geriatricians and General Practitioners | Y Y |
| 12. | Neurology[13] | 2017 | Association of British Neurologists | Y |
| 13. | Human nutrition[17] | 2001 | National Nutrition Task Force | Y |
| 14. | Obstetrics and Gynaecology[7] | 2015 | Royal College of Obstetricians and Gynaecologists | Y |
| 15. | Ophthalmology[28] | 2015 | Royal College of Ophthalmologists | Y |
| 16. | Oncology[26] | 2014 | The Royal College of Radiologists | Y |
| 17. | Paediatrics[6] | 2015 | Royal College of Paediatrics and Child Health | Y |
| 18.A B | Palliative care[11] Palliative medicine[12] | 2014 2008 | Scottish Palliative Medicine Curriculum Development Group Association for Palliative Medicine | Y Y |
| 19. | Pathology[16] | 2014 | The Royal College of Pathologists | Y |
| 20. | Patient safety[36] | 2011 | WHO | Y |
| 21. | Psychiatry[8] | 2011 | Royal College of Psychiatrists | Y |
| 22. | Public health[29] | 2014 | Faculty of Public Health | Y |
| 23. | Radiology[27] | 2017 | Royal College of Radiologists | Y |
| 24. | Sociology[31] | 2016 | Behavioural and Social Sciences Teaching in Medicine Sociology Steering Group | Y |
| 25. | Surgery[9] | 2015 | Royal College of Surgeons | Y |
| 26.A B | Urology[32] Urology syllabus[33] | 2002 2012 | Journal article in Urology British Association of Urological Surgeons | Unclear Unclear |

ENT, ear, nose and throat; GMC, General Medical Council; WHO, World Health Organization.

### Drivers for development of core curricula
#### Patient safety
Patient safety was a common theme in the development of core curricula for clinical specialties, foundation subjects, in addition to patient expectations of a graduate in terms of medical ethics and law.[15 18 21–23 27] WHO

(World Health Organization) has published the multiprofessional patient safety curriculum guide to aid teaching of patient safety in universities and schools, including medicine.[36] The ACUTE (Acute Care Undergraduate Teaching) initiative identified the core set of competencies for resuscitation and care of acutely ill patients that all UG trainees require to be competent in at the point of graduation.[10] This was designed to address the suboptimal care of the critically ill in hospital where contributory factors included recognition of critical illness, lack of knowledge, a lack of supervision, failure to seek advice and poor communication.[10] Similarly, the British Pharmacology Society prepared a curriculum to standardise the learning objectives and recommended introduction of prescribing safety assessments in all medical schools, in alignment with GMC's outcomes.[30]

### Disease burden and needs of Society

In the UK, cancer affects one in three people.[26] The Royal College of Radiologists, who developed the UG radiology curriculum, also developed the non-surgical oncology curriculum. With collaboration from the Royal College of Physicians' Joint Collegiate Council for Oncology (JCCO), these UG curricula in a joint approach aim to address competencies in a multidisciplinary setting towards patients affected with cancer and their needs.[26 27]

Societal requirements of an ageing, frail population, with complex problems related to independence and mobility were the important factors for the development of a core curricula for geriatric medicine and the musculoskeletal system.[22–24]

The National Health Service of the UK is underpinned by GPs, who remain the first point of contact for patients and provide long-term continuity of care. To provide a sustainable service to patients, UK needs approximately half of medical school graduates to pursue a career in family medicine or general practice.[25] In 2018, the Royal College of General Practitioners with the Society for Academic Primary Care collaborated to develop the GP curriculum as guidance to UK medical schools on the design and delivery of teaching of general practice to help bridge the gap between societal demands and needs.[25]

Public healthcare prevention and promotion, human nutrition and its relevance to health and disease also had specific UG curricula.[17 25]

### Lack of representation and inadequate preparedness amongst graduates

Multiple specialties, through national surveys across UK medical schools, felt a lack of adequate representation, teaching or assessments in medical school curricula.[13 14 32–34 37] These concerns were echoed by feelings of inadequacy among junior doctors and GPs due to a lack of UG training in some specialties, and some of these specialties subsequently developed their own curricula.[13 38 39] This is particularly seen in specific core curricula like ear, nose and throat (ENT) and dermatology, where related diseases and

conditions are common presentations in the UK primary care setting.[14 34] This has led such specialties to develop specialty-specific learning objectives, defining minimum knowledge and skills to be attained.

### Curriculum: overlaps and gaps

The GMC, based on a review of curriculum theories, its relevance and context with regard to medical education and its stakeholders, that is, students, teachers and regulators, defines 'a curriculum as a statement of the intended outcomes, encompassing content, teaching, learning and assessment methods, feedback and supervision as part of the educational programme'.[40]

This scoping review found that the Royal Colleges of the UK, such as the Royal College of Surgeons (RCS) and Obstetrics and Gynaecology, covered the curricular requirements comprehensively by highlighting content and how to achieve the learning objectives with structure and assessment methods.[7 9]

However, many clinical and foundation subjects suggested medical schools determine how best to incorporate content within their medical course to allow flexibility in their implementation, delivery and assessment. Hence, they were focused primarily on syllabus content.[13 14 28 34]

Despite representation of surgical subspecialties like ENT and urology in the RCS curriculum, both subspecialties developed specific outcomes for medical school curricula.[32–34] Overlapping of core content was also seen between more established curricula like geriatrics[2] and psychiatry.[8] Interestingly, there was no specific UG curriculum for medicine even though the RCP sets out postgraduate standards for specialty physicians in the UK.

## DISCUSSION

We used a structured literature search to identify publications, which described core curriculum recommendations for a specialty or subject for the UK medical UGs. We identified common factors and themes, and bodies who have developed a core curriculum for the UK medical schools, grouping specialties into clinical, foundation and professionalism-related subjects.

Medical schools have an enormous task of continuously responding to changes in the clinical practice, keeping the curriculum in context and ensuring that students are not overburdened by content. From our review of the available literature, we demonstrated that most of the available specialty-specific core curricula had been developed through international and national expert consensus, advising on minimum standards on core knowledge and skills to be achieved by the medical graduates and aligned themselves to the Outcomes for graduates document. The 'core knowledge' in specialty-specific curricula could help to develop minimum standards and provide quality assurance for both generic and specialty-specific areas across all the UK medical schools. The

drivers to curriculum development included patient safety, disease burden and needs of society, and lack of representation or inadequate preparedness among graduates. We also identified areas where there were gaps or overlaps in some specialties' curricula. Though the GMC recommends a 'core curriculum' for UG medical education, medical educators have found it challenging to define what is meant by core, what needs to be included and its relevance to the bigger context of competence for graduating medical students.[1–3]

While medical schools align their curricula to the generic 'Outcomes for graduates', specialty representation and mapping of burden of disease in curricula is challenging, and how individual providers of medical education map curriculum content, learning outcomes and assessments for specialties is not explicit.[41 42] As Voss et al have noted, 'Improving patient safety and quality in health care is one of medicine's most pressing challenges'.[43] The GMC has also highlighted these areas in the new Outcomes for graduates.[3]

In the review, it was observed that there was duplication and overlapping of some specialties and subjects with their core curricular recommendations. For this review, curricula developed on topics within specialties were excluded. For example, a curriculum for delirium has been developed, which is a topic covered within psychiatry and geriatrics specialty.[44] Similarly, recently, an ultrasound curriculum[45] for medical students has been developed in addition to the national UG curriculum for radiology. Perhaps necessarily, certain clinical conditions demand a multidisciplinary approach. An integrated approach to combine specialties would help to reduce content overload, while keeping teaching and learning contextual. The introduction of longitudinal integrated clerkship in medical schools is looking to address this issue.[42] Further adoption of interprofessional education can help to change attitudes and behaviours and provide a sustainable model to accommodate large numbers of students.[46 47] For example, the non-surgical oncology and palliative care national curricula could be integrated and specialist nurses involved in helping medical students understand different aspects of cancer and end-of-life care.[11 12 26]

There were several drivers of curriculum development. For example, in 2016, the UK government announced strategies to fund, develop and grow the general practice workforce, to meet the increasing demands of an ageing population with complex needs and chronic disease burden.[48] The new UG curriculum for general practice has been developed to complement postgraduate curriculum reflecting how curricula have evolved in relation to healthcare service and societal needs.[25] Furthermore, the introduction of a UK-wide MLA in 2022 may influence what medical schools teach and provide opportunities to keep curricular content relevant and inclusive.

Thus, by examining the evidence for need to change or update medical curricula, and through reflection on content and process, Mezirow's concept of transformative learning could be applied by the medical educators in curricular development.[49] For instance, in Scotland, the Scottish Deans' Medical Curriculum Group discusses and helps coordinate the development, delivery and evaluation UG curricula. Their aim is to ensure that graduates from each school in Scotland are of an equivalent standard. Their review of the curriculum has also been informed by frameworks on specialist subjects like acute and emergency medicine, dermatology, neurology, palliative care, pharmacology and sexual health.[50]

The curricula developed for foundation and professionalism-related subjects recommended their components be implemented by integration both horizontally and vertically throughout the medical course. These subjects, relevant and applicable to all clinical specialties, could be included in generic outcomes for the specialties and form a framework through the UG years.

It is vital for UG curricula to evolve by keeping abreast with changing clinical needs and advances in diagnostics, investigative and management strategies.

With limited availability of resources and time, the curricula could be split into essential, desirable and nice to know categories to help prioritise areas of teaching and learning. The use of technological advances (eg, use and sharing of e-learning resources/webinars) could help to provide flexibility and autonomy in how these curricula could be implemented and delivered, if followed.

### Limitations of the review

We conducted a comprehensive search for all relevant articles for this study. However, as with limitations seen in scoping reviews, articles or studies pertinent to the review question may have been unintentionally omitted.[3] The level of evidence for the quality of the developed specialty specific core curricula could not be graded in this search, as done with systematic reviews. However, most curricula were developed with expert consensus and use of a Delphi study, suggesting that efforts were made to establish 'best practice' and core curricular outcomes. The curricular components for individual specialties were variable, making comparisons difficult.

We were nevertheless able to establish common themes and drivers for development of the UG core curricula for the UK medical UGs. In addition, the study helped to identify gaps in specialty-specific curricula like the absence of an overarching RCP curricula, as well as the heterogeneity of the curricular designs between specialties.

### CONCLUSIONS

This literature review highlights the status of specialty-specific core curricula for medical graduates in the UK. This is the first study to provide a comprehensive overview of the curricula developed by specialties and their recommendations on knowledge and skills expected of all UK medical graduates, in alignment with GMC. The review establishes common themes and drivers for development of core curricula for UK medical UGs. The study

identifies gaps in development of specific UG curricula as well as the heterogeneity of the curricular designs between specialties and subjects. It would be interesting to see how medical schools are implementing and assessing these curricula and how that is likely to affect future training.

**Acknowledgements** Rakesh Patel, Clinical Associate Professor in Medical Education and Honorary Consultant Nephrologist, Faculty of Medicine and Health Sciences, University of Nottingham for guidance on the review question Lisa Lawrence, Clinical librarian, Burton and Derby Teaching Hospitals NHS Foundation Trust, Derby for conducting the searches for the review Prof Cathy Bennett, Royal College of Surgeons in Ireland (RCSI), Office of Research and Innovation. Manuscript revision. Some of the work has been previously presented as oral presentation at: British Association of Dermatologist (BAD) Teachers of Undergraduate Dermatology Annual Meeting on 6th March 2018 Association for the Study of Medical Education (ASME) Annual Scientific Meeting on 12 July 2018.

**Contributors** All authors have made substantial contributions to the planning, conduct, analyses and interpretation of findings, and reporting of the work described in the article, and have agreed to be accountable for all aspects of the work, its accuracy and integrity. MS is responsible for the overall content as guarantor. MS, RM and GAD with RP formulated the review question. MS screened the titles and abstracts of retrieved citations and MS and RM assessed the full text. MS, RM and GAD resolved any ambiguities about inclusion criteria. RM and MS devised the data extraction proforma. MS extracted data and discussed data items with RM and GAD. MS contacted study authors for further information. RM and MS identified common themes, MS conducted the narrative synthesis with RM and GAD. MS wrote the first draft of the manuscript and revised and amended it with RM and GAD who also approved the final version to be published. MS is the corresponding author. The corresponding author attests that all listed authors meet authorship criteria and that no others meeting the criteria have been omitted.

**Funding** The authors have not declared a specific grant for this research from any funding agency in the public, commercial or not-for-profit sectors.

**Competing interests** None declared.

**Patient consent for publication** Not required.

**Provenance and peer review** Not commissioned; externally peer reviewed.

**Data availability statement** All data relevant to the study are included in the article or uploaded as online supplementary information.

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
