## [Reviewer comments · BMJ Open]

ARTICLE DETAILS

TITLE (PROVISIONAL)	DO WE NEED A CORE-CURRICULUM FOR MEDICAL STUDENTS? A SCOPING REVIEW
AUTHORS	Sharma, Maulina; Murphy, Ruth; Doody, Gillian

VERSION 1 – REVIEW

REVIEWER	Joselina Barbosa Faculty of Medicine of the University of Porto, Portugal
REVIEW RETURNED	12-Nov-2018

GENERAL COMMENTS	The introduction is too generic. It should be explained the importance of the subject, for example, what is the purpose of the documents created by the GMC and what is the role of the medical generalist. Some of these are in the discussion section. The Introduction has no main objective. Why do the specialities expressed concern about medical curricula? I think the discussion does not fully meet the main objective of the study. I expected to be discussed how medical schools link GMC standards and the specialty specific core curricula. Also, should be discussed how to implement a curriculum taking into account: the GMC standards, the specialty specific core curricula, the gaps and drivers found in the results and the medical generalists (as described in the Introduction section.)
---

REVIEWER	Joachim Graf University Hospital Tübingen, Germany
REVIEW RETURNED	02-Dec-2018

GENERAL COMMENTS	A well written paper using the Scoping reviews methodology. It would be nice if you would add one or two illustrations to visualize the results of the literature review.
---

VERSION 1 – AUTHOR RESPONSE

Reviewer: 1

Reviewer Name: Joselina Barbosa

Institution and Country: Faculty of Medicine of the University of Porto, Portugal

Please state any competing interests or state 'None declared': None declared

The introduction is too generic. It should be explained the importance of the subject, for example, what is the purpose of the documents created by the GMC and what is the role of the medical generalist. Some of these are in the discussion section. (Thank you. We have addressed the reviewer's comments)

The Introduction has no main objective. (Thank you. We have now clearly stated our objective within the introduction)

Why do the specialities expressed concern about medical curricula? (Thank you. We have addressed this on page 4, paragraph 1).

I think the discussion does not fully meet the main objective of the study. I expected to be discussed how medical schools link GMC standards and the specialty specific core curricula. Also, should be discussed how to implement a curriculum taking into account: the GMC standards, the specialty specific core curricula, the gaps and drivers found in the results and the medical generalists (as described in the Introduction section.) (We thank the reviewer for this comment. We have revised and reorganised the discussion completely so it is aligned to the review question, aims and objectives. Specifically, we discuss how medical schools currently link to GMC standards. However, specialty specific curricula are not explicitly standardised amongst medical schools. Lack of knowledge about this issue provided the rationale for our research)

Reviewer: 2

Reviewer Name: Joachim Graf

Institution and Country: University Hospital Tübingen, Germany

Please state any competing interests or state 'None declared': None declared

A well written paper using the Scoping reviews methodology. (We thank the reviewer for their kind comments.)

It would be nice if you would add one or two illustrations to visualize the results of the literature review. (If we have understood correctly, we have provided a PRIMA diagram in figure 1, added the search strategy in full in appendix, and table 1 provides the collated information for the individual studies. We have reorganised and added texts to the methods and results section to aid the reader.)

FORMATTING AMENDMENTS (if any)

Required amendments will be listed here; please include these changes in your revised version:

1. Kindly embed your Article Summary section in your main document file. (the article summary is embedded)

2. - Please remove all your figures in your main document and upload each of them separately under file designation 'Image' (except tables and please ensure that Figures are of better quality or not pix-elated when zoom in). NOTE: They can be in TIFF, JPG or PDF format and make sure that they have a resolution of at least 300 dpi. Figures in DOCUMENT, EXCEL and POWER POINT

format are not acceptable. (Thank you. We have now attached the PRISMA figure 1 as a separate image file in PDF format)

3. - Please provide a more detailed contributorship statement. It needs to mention all the names/initials of authors along with their specific contribution/participation for the article. This should list each author's contribution to the paper according to the ICMJE guidelines for authorship. This should be stating how each author contributed to the article. It should discuss on the planning, conduct and reporting of the work in your paper. You may also consider the conception and design, acquisition of data or analysis and interpretation of data, etc. .(We have included this now, thank you)

VERSION 2 – REVIEW

REVIEWER	Joselina Barbosa Faculty of Medicine of the University of Porto
REVIEW RETURNED	10-Apr-2019
GENERAL COMMENTS	The article has improved. The conclusion is very general. For example, for those who read only the abstract does not realize what the study adds